# Patients’ Satisfaction and Oral Health-Related Quality of Life of Edentulous Patients Using Conventional Complete Dentures and Implant-Retained Overdentures in Saudi Arabia

**DOI:** 10.3390/ijerph19010557

**Published:** 2022-01-04

**Authors:** Salwa Omar Bajunaid, Abdullah S. Alshahrani, Ahad A. Aldosari, Atheer N. Almojel, Rehab S. Alanazi, Tala M. Alsulaim, Syed Rashid Habib

**Affiliations:** Department of Prosthetic Sciences, College of Dentistry, King Saud University, Riyadh 11545, Saudi Arabia; asalshahrani@ksu.edu.sa (A.S.A.); ahad.aldosari@gmail.com (A.A.A.); atheer.n.i.m@gmail.com (A.N.A.); reehhaab977@gmail.com (R.S.A.); talaalsu@gmail.com (T.M.A.); syhabib@ksu.edu.sa (S.R.H.)

**Keywords:** implant, overdenture, quality of life, satisfaction

## Abstract

The aim of this study was to compare patients’ satisfaction and quality of life (QoL) when using implant overdentures vs. conventional dentures. The QoL of Saudi patients who were provided mandibular implant-supported overdentures was assessed using a structured questionnaire. Overall satisfaction; ability to speak, eat, and chew food; comfort; esthetics; stability; and satisfaction of general oral health were measured. A total of 48.3% vs. 6.9% were, overall, very satisfied with their implant overdentures and conventional dentures, respectively. A total of 37.9% of the patients were very satisfied regarding speaking with their implant overdentures vs. 17.2% with conventional dentures. Furthermore, 41.4% were very comfortable with their implant overdentures vs. 5.2% were very comfortable using conventional dentures. However, only 15.5% were very satisfied with the esthetics of the conventional dentures compared to 43.1% being satisfied with implant overdentures. Only 1.7% were very satisfied with the stability of conventional dentures vs. 44.8% being satisfied using implant overdentures. About 8.6% of the candidates were very satisfied regarding chewing food with conventional dentures vs. 36.2% being very satisfied using implant overdentures. Approximately 10.3% were very satisfied with their general oral health using conventional dentures compared to 29.3% being very satisfied using implant overdentures. Mandibular implant overdentures had a strong impact on patients’ quality of life over conventional complete dentures and should be considered the minimum standard of care provided to completely edentulous patients.

## 1. Introduction

Rehabilitation of edentulous patients with removable dentures is a long-standing procedure that was first prescribed by Purmann in 1684 [1]. It exerts a great positive impact on people’s daily life as dentures help restore an individual’s esthetics, normal oral function, and the ability to socialize [2]. Shah et al. found that approximately 58% of edentulous people felt upset and experienced social isolation due to functional and appearance problems [3].

Conventional removable complete dentures (which use soft tissue as means for support, retention, and stability) still fail to provide superior function and comfort for patients. In the 1990s, and after the introduction of endosseous dental implants and their use to retain and/or support removable prostheses, many authors started to evaluate the differences between conventional dentures and implant-retained overdentures [4,5]. Awad et al. (2014) and Kodama et al. (2016) found that, when compared with conventional dentures, mandibular two-implant overdentures significantly improved patient satisfaction and oral health-related quality of life (OHRQL) for edentulous patients [6,7].

A study that was conducted by Cibirka in 1997 showed a significant improvement in self-image among the implant group along with better function and comfort [8]. Another study determined that in addition to a significant level of satisfaction, 97% of the patients reported that their implants were worth the trouble, and 89% would repeat the procedure if they could [9].

Several reports on satisfaction with implant-supported prostheses among edentulous patients have been published, showing a significant improvement in psychological self-image among the implant group along with better function and comfort [10,11].

In 2002, the McGill consensus declared that “two-implant mandibular overdentures should replace mandibular conventional dentures as the minimum standard of care for completely edentulous patients” [12]. This was followed by the York statement in 2009, which concluded that “a substantial body of evidence is now available demonstrating that patients’ satisfaction and quality of life (QOL) with implant-supported mandibular overdentures (ISOD) is significantly greater than for conventional dentures. It is the minimum standard that should be sufficient for most people” [13]. An umbrella systematic review conducted by Sharka et al. demonstrated the superiority of using implant overdentures compared to conventional complete dentures on the oral health quality of life (OHQoL) and patients’ satisfaction outcomes [14]. More recently, several studies have shown that the QoL of patients with conventional dentures is inferior to those with implant-supported overdentures [15,16].

Implant-supported overdentures have different types of attachments, and their use is affected by multiple factors including the practitioner’s choice and preference. There have been studies that advocate the use of bar attachments over stud attachments, as reported by Boven (2019) [17]; however, both provide expected outcomes given that the choice of attachment is selected correctly [18]. Another study by Matthys (2019) reported a 5-year implant survival rate of 98.7%, irrespective of the type of attachment system used [19].

Many forms and scales exist by which to estimate patients’ quality of life (QoL), such as the Oral Health Impact Profile (OHIP) and Oral Health-Related Quality of Life (OHRQoL). In terms of prosthodontically-related evaluation, an assessment of prosthesis by McGill University in 2001 evaluated both patients’ satisfaction and QoL.

Although several studies have established that implant-retained overdentures are better than conventional dentures in terms of stability, retention, comfort, and masticatory function, currently, there are no studies performed on a Saudi Arabian population. Thus, the aim of this survey was to compare patients’ satisfaction and QoL when using conventional complete dentures vs. implant-supported overdentures. The results of this study may help implement implant-retained overdentures as a first choice in the treatment plan of edentulous patients in Saudi Arabia.

## 2. Materials and Methods

Fifty-eight edentulous individuals who were provided mandibular implant-supported overdentures in the Kingdom of Saudi Arabia were included. Quality of life (QoL) was assessed by using a structured questionnaire, self-filled by participants who were selected by convenience sampling. Each respondent was ensured of their confidentiality and anonymity, with sufficient time given to fill the questionnaire. Illiterate participants were interviewed by questionnaire distributors or any of the caregivers.

The inclusion criteria included edentulous patients who use mandibular two-implant-supported overdentures for more than 3 months and those who had previous experience using conventional complete dentures. The exclusion criteria included candidates without previous experience with conventional complete denture use, those who do not use their implant-supported overdenture, and patients with maxillary implant-retained overdentures or with tooth-supported overdentures. Moreover, patients with less than 3 months of having an implant-supported overdenture were not questioned because the period of time was insufficient to measure QoL after receiving the treatment.

The study protocol was approved by both the College of Dentistry Research Center (CDRC) and the Institutional Review Board (IRB) at King Saud University (KSU), Saudi Arabia. The study was performed in accordance with the ethical principles of the Helsinki Declaration and KSU policies and guidelines for research. All of the patients’ information from the included cases was confidential and kept anonymous. Participation was voluntary; the nature of the study was explained at the beginning of the questionnaire; no attempt was made to collect the participants’ information.

The questionnaire used was an Arabic translation of the McGill University 2001 Assessment of Prosthesis [20]. It contained the following three parts: the first part included the demographic data of the patients (gender, age, and provenance) along with the duration for which the patients had been using the lower implant-retained overdenture. The second part included 17 questions, which focused on comparing conventional complete dentures and implant-retained overdentures in the following areas: general satisfaction, ease of cleaning, ability to speak, comfort, esthetics, stability, ability to chew different types of foods, function, and their general oral condition. Patients responded on a 5-point Likert scale. The third part included a single, open-ended question, and the patients could provide their opinion on implant-retained overdentures.

The data were analyzed using the SPSS 26.0 statistical software (IBM Inc., Chicago, IL, USA). Descriptive statistics (frequencies and percentages) were used to describe the categorical study and outcome variables. A non-parametric statistical test (Wilcoxon signed-rank) was used to compare the repeated 5-point scale responses of the study subjects to compare the satisfaction levels with the use of conventional dentures vs. implant overdentures. A *p*-value of ≤0.05 was used to report the statistical significance of results.

## 3. Results

Out of 58 study subjects who received implant-retained overdentures, 33 (56.9%) were male subjects; more than 80% of them were over 45 years of age. A total of 41 (70.7%) of the subjects were living in the central region of Saudi Arabia, and 29 (50%) had been using implant overdentures for more than 2 years. Approximately two thirds of all subjects (65.5%) had the perception that their oral health negatively affected their overall general health (Table 1).

The quality of life was assessed using 13 items that measured the subjects’ level of satisfaction with conventional dentures and implant overdentures. The 5-point scale (1 = very satisfied; 2 = satisfied; 3 = neutral; 4 = not satisfied; 5 = not satisfied at all) was used to assess patients’ satisfaction levels.

The assessment of the ‘overall satisfaction’ with conventional dentures showed that only 6.9% and 1.7% of the patients were very satisfied and satisfied, respectively. However, for implant overdenture, 48.3% and 1.7% of them were very satisfied and satisfied, respectively. The assessment of ‘ease of cleaning mouth and denture’ showed that 29.3% and 41.4% of the subjects using conventional denture were very satisfied and satisfied, respectively; 34.5% and 43.1% of those using implant overdentures were very satisfied and satisfied, respectively. For the item *‘ability to speak’*, 17.2% and 19% of the patients were very satisfied and satisfied with their conventional denture, respectively; 37.9% and 36.2% of them were very satisfied and satisfied using implant overdentures, respectively. For the *‘comfort’* item, only 5.2% and 25.9% of the patients were very satisfied and satisfied, respectively, with the conventional denture, whereas a higher proportion (41.4% and 31%) of the patients were very satisfied and satisfied with the implant overdenture. The assessment of *‘satisfaction with esthetics’* showed that 15.5% and 5.2% of the subjects were very satisfied and satisfied, respectively, compared to 43.1% and 3.4% of the subjects using implant overdenture. For the *‘satisfaction with stability’* item, only 1.7% and 3.4% of the candidates were very satisfied and satisfied, respectively, compared with the proportion of 44.8% and 1.7% of very satisfied and satisfied patients using implant overdentures. The assessment of the *‘difficulty in chewing food’* showed that only 8.6% and 12.1% of the study subjects expressed that they were very satisfied and satisfied with the use of conventional dentures compared to 36.2% and 31% of the patients who were very satisfied and satisfied with the use of implant overdentures, respectively. For the item *‘difficulty in chewing and eating white bread’*, only 3.4% and 17.2% stated that they were very satisfied and satisfied, respectively, while using conventional dentures compared to a higher proportion of 31% and 34.5% who mentioned that they were very satisfied and satisfied with the use of implant overdentures, respectively. For the item *‘difficulty in chewing and eating dates’*, only 6.9% and 17.2% stated that they were very satisfied and satisfied, respectively, while using conventional dentures compared to a higher proportion of 27.6% and 31% of the subjects who mentioned that they were very satisfied and satisfied with their implant overdentures, respectively. For the item *‘difficulty in chewing and eating cheese’*, approximately 10.3% and 20.7% were very satisfied and satisfied, respectively, while using conventional dentures compared to a higher proportion of 29.3% and 39.7% who were very satisfied and satisfied with their implant overdentures, respectively. For the item *‘difficulty in chewing* and *eating meat’*, only 1.7% and 13.8% of the subjects were very satisfied and satisfied, respectively, while using conventional dentures compared to a higher proportion of 24.1% and 34.5% of them who were very satisfied and satisfied with the use of implant overdenture, respectively. For the item *‘difficulty in chewing* and *eating apple’*, only 1.7% and 15.5% of the patients were very satisfied and satisfied, respectively, while using conventional dentures compared to a higher proportion of 17.2% and 32.8% of them who were very satisfied and satisfied with the use of implant overdentures, respectively. The assessment of satisfaction item *‘difficulty in chewing* and *eating lettuce’* showed that only 3.4% and 15.5% of the patients were very satisfied and satisfied, respectively, while using conventional dentures compared to a higher proportion of 32.8% and 22.4% who stated that they were very satisfied and satisfied with the use of implant overdentures, respectively.

For the item of ‘satisfaction of general oral health’, approximately 10.3% and 1.7% of the patients were very satisfied and satisfied, respectively, with conventional dentures compared to 29.3% and 3.4% of the patients who were very satisfied and satisfied with the use of implant overdentures (Table 2).

Table 3 shows the mean negative and positive ranks, which were derived from the responses of 5-point satisfaction levels for 13 assessment items so as to compare the use of conventional dentures and implant overdentures. The Wilcoxon signed-rank test statistics and the corresponding *p*-values indicate a highly statistically significant difference in the responses of study subjects to 12 items regarding their use of conventional dentures and implant overdentures. Specifically, the responses ‘very satisfied and satisfied’ were more frequent for implant overdentures compared to similar responses for conventional dentures. In addition, the responses ‘not satisfied and not satisfied at all’ were significantly less frequent for implant overdentures compared to similar responses for conventional dentures. For the *‘ease of cleaning mouth and denture’* item, there was no statistically significant difference in the responses regarding the use of conventional and implant overdentures (*p* = 0.583) (Table 3).

The study subjects were asked *‘Do you think that your oral health has a negative impact on your general health? If yes, why?*’. This open-ended question revealed responses such as: the mouth is a reservoir for bacteria and tooth decay; not able to chew food well; the mouth is the means for nourishment for the body; difficulty in eating and want to eat using teeth and be able to talk; it affects both physical health and psychological aspects; cannot eat unless chewing the food well; any sores in the mouth will affect eating and negatively affect the entire body; if food is not chewed well, then stomachache and constipation will occur; with improper chewing of food, indigestion will occur. Some subjects experienced ‘embarrassment while speaking’, and if one’s oral condition is bad, this affects the gingiva and the jawbone, which leads to problems in the whole body.

Table 4 shows the open-ended responses of the study subjects, which indicate different reasons for the patients’ preference of using implant overdentures vs. conventional dentures.

## 4. Discussion

This study was conducted to determine the effect of using mandibular implant-retained overdentures on Saudi patients’ satisfaction and QoL compared to using their previous conventional complete dentures. Several studies and consensus statements have established that implant-retained overdentures are a better option for the treatment of mandibular edentulism in terms of retention, stability, and overall QoL [12,13,14]. A systematic review by Kutkut et al. has shown that mandibular implant-retained overdentures are superior to conventional complete dentures because patients have high levels of satisfaction, QoL, and function [5]. The results of this current study agree with the abovementioned results, which suggest that patients experienced great improvements in their overall QoL with the transition from conventional complete dentures to implant-retained overdentures.

Regarding the ease of cleaning the mouth and dentures, patients reported no significant differences between the two treatment options; however, this contradicts a study performed by Martín-Ares in 2016, who determined that patients with implant-retained overdentures reported better satisfaction with their oral hygiene compared to patients using conventional complete dentures [21].

The patients in this study reported significantly improved satisfaction with speaking, comfort, esthetics, and the stability of the implant-retained overdentures; these results are consistent with those of Melas et al. (2001) and Heydecke et al. (2005), who reported that patients experienced less difficulty in performing daily activities such as speaking, smiling, and eating after the use of implant-retained overdentures [22,23]. Fernandez-Estevan et al. (2015) and Mishra et al. (2019) also reported that retention, stability, comfort, speech, and chewing efficiency improved drastically with enhanced patient satisfaction and a better quality of life (QoL) when using implant overdentures vs. conventional complete dentures [15,24]. However, a study by Assunção et al. in 2007 showed that despite the enhanced stability of implant-retained overdentures, there was no significant difference between treatments in terms of QoL, overall satisfaction, ability to chew, comfort, or esthetics [25].

The present study suggests that mandibular implant overdentures considerably affect the ability to chew different types of food. A study by Feine et al. in 2002 documented that many patients wearing conventional dentures complained that they could not eat many types of food, particularly hard food. This issue motivated them to change their diets and made their nutrition worse than that of individuals with natural teeth [12]. This study showed that patients were more satisfied with their ability to chew different types of food with less difficulty after using mandibular implant overdentures. These results agree with those of Awad et al. (2012), Boven et al. (2015), and Sharma et al. (2017), who determined that implant-stabilized overdentures have a better ability to improve the chewing efficiency and increase maximum biting force than conventional complete dentures [26,27,28].

The effect of general oral health on patients’ quality of life (OHRQoL) was assessed, and this revealed that the level of patients’ satisfaction improved with the use of mandibular implant overdentures compared to conventional dentures. This is in agreement with a study carried out by Heydecke and colleagues who found that mandibular two-implant overdentures provided better OHQoL and improved patients’ general health [29]. This effect of mandibular two-implant overdentures was found to be stable over a two-year assessment period [30].

The last question in this study was an open-ended question, which was used to assess the negative impact of oral health on general health. Some patients think that there is a relationship between not chewing food properly and digestive issues; consequently, they believe that their general health will be negatively affected. These findings are in agreement with a study conducted by De Andrade et al., which showed that among elderly Brazilians, a high degree of negative impact of oral health on QoL was associated with patients’ general health [31]. Furthermore, a nonsystematic review concluded that the oral health status of the geriatric population directly affected the individual’s general quality of life and their well-being [32].

## 5. Conclusions

The results of this retrospective study showed that mandibular implant-retained overdentures greatly improved the patients’ quality of life (QoL) compared with mandibular conventional complete dentures. The use of two dental implants to retain mandibular overdentures significantly improved denture stability and retention and thus positively enhanced the patients’ comfort, esthetics, and satisfaction while speaking compared with the use of conventional complete dentures. Moreover, the implant overdentures enabled patients to efficiently chew different types of food and hence improved their nutrition and well-being. Patients who participated in this study observed that their oral health has a direct impact on their general health.

## Figures and Tables

**Table 1 ijerph-19-00557-t001:** Characteristics of study subjects (*n* = 58).

Characteristics	No. (%)
**Age groups**	
<30	2 (3.4)
31–45	9 (15.5)
46–60	19 (32.8)
61–75	25 (43.1)
>75	3 (5.2)
**Sex**	
Male	33 (56.9)
Female	25 (43.1)
**Region**	
Central	41 (70.7)
Western	12 (20.7)
Eastern	4 6.9)
Southern	1 (1.7)
**Time since use of implant overdenture**	
3 months	7 (12.1)
6 months	7 (12.1)
12 months	15 (25.9)
>2 years	29 (50.0)
**Do you think your general oral health has a negative impact on your general health?**	
Yes	38 (65.5)
No	20 (34.5)

**Table 2 ijerph-19-00557-t002:** Distribution of satisfaction level responses towards using conventional implant dentures and implant overdentures (*n* = 58).

Items	Conventional Denture No. (%)					Overdenture No. (%)				
	1	2	3	4	5	1	2	3	4	5
Satisfaction using denture	4 (6.9)	1 (1.7)	19 (32.8)	21 (36.2)	13 (22.4)	28 (48.3)	1 (1.7)	18 (31)	7 (12.1)	4 (6.9)
Ease of cleaning mouth and denture	17 (29.3)	24 (41.4)	11 (19)	5 (8.6)	1 (1.7)	20 (34.5)	25 (43.1)	6 (10.3)	6 (10.3)	1 (1.7)
Ability to speak	10 (17.2)	11 (19)	16 (27.6)	12 (20.7)	9 (15.5)	22 (37.9)	21 (36.2)	7 (12.1)	4 (6.9)	4 (6.9)
Comfort	3 (5.2)	15 (25.9)	6 (10.3)	21 (36.2)	13 (22.4)	24 (41.4)	18 (31.0)	4 (6.9)	9 (15.5)	3 (5.2)
Satisfaction with esthetics	9 (15.5)	3 (5.2)	27 (46.6)	13 (22.4)	6 (10.3)	25 (43.1)	2 (3.4)	24 (41.4)	4 (6.9)	3 (5.2)
Satisfaction with stability	1 (1.7)	2 (3.4)	18 (31.0)	19 (32.8)	18 (31.0)	26 (44.8)	1 (1.7)	20 (34.5)	7 (12.1)	4 (6.9)
Difficulty chewing food	5 (8.6)	7 (12.1)	11 (19)	19 (32.8)	16 (27.6)	21 (36.2)	18 (31.0)	7 (12.1)	6 (10.3)	6 (10.3)
Difficulty chewing and eating white bread	2 (3.4)	10 (17.2)	14 (24.1)	17 (29.3)	15 (25.9)	18 (31)	20 (34.5)	7 (12.1)	6 (10.3)	7 (12)
Difficulty chewing and eating dates	4 (6.9)	10 (17.2)	10 (17.2)	18 (31)	16 (27.5)	16 (27.6)	18 (31)	12 (20.7)	6 (10.3)	6 (10.3)
Difficulty chewing and eating cheese	6 (10.3)	12 (20.7)	14 (24.1)	15 (25.9)	11 (18.9)	17 (29.3)	23 (39.7)	6 (10.3)	6 (10.3)	6 (10.3)
Difficulty chewing and eating meat	1 (1.7)	8 (13.8)	12 (20.7)	17 (29.3)	20 (34.5)	14 (24.1)	20 (34.5)	8 (13.8)	11 (19)	5 (8.6)
Difficulty chewing and eating apple	1 (1.7)	9 (15.5)	6 (10.3)	22 (37.9)	20 (34.5)	10 (17.2)	19 (32.8)	11 (19)	9 (15.5)	9 (15.5)
Difficulty chewing and eating lettuce	2 (3.4)	9 (15.5)	16 (27.6)	13 (22.4)	18 (31)	19 (32.8)	13 (22.4)	12 (20.7)	6 (10.3)	8 (13.7)
Satisfaction of general oral health	6 (10.3)	1 (1.7)	25 (43.1)	17 (29.3)	9 (15.5)	17 (29.3)	2 (3.4)	29 (50)	8 (13.8)	2 (3.4)

1 = very satisfied; 2 = satisfied; 3 = neutral; 4 = not satisfied; 5 = not satisfied at all.

**Table 3 ijerph-19-00557-t003:** Comparison of mean ranks of satisfaction level responses between conventional dentures and implant overdentures.

Items	Mean Negative Ranks	Mean Positive Ranks	Wilcoxon Signed-Rank Test Statistic	*p*-Value
Satisfaction using denture	20.45	13.2	−4.455	<0.001
Ease of cleaning mouth and denture	11	12.22	−0.548	0.583
Difficulty in talking	16.4	17.5	−4.013	<0.001
Comfort	19.16	9	−4.654	<0.001
Satisfaction with esthetics	11.38	3.5	−3.925	<0.001
Satisfaction with stability	19.29	5	−5.115	<0.001
Difficulty chewing food	21.17	8.5	−4.81	<0.001
Difficulty chewing and eating white bread	19.95	9.5	−4.525	<0.001
Difficulty chewing and eating dates	19.25	10.5	−4.34	<0.001
Difficulty chewing and eating cheese	16.35	10	−4.004	<0.001
Difficulty chewing and eating meat	21.31	9.83	−4.555	<0.001
Difficulty chewing and eating apple	19.47	9.2	−4.45	<0.001
Difficulty chewing and eating lettuce	18.55	8.3	−4.306	<0.001
Satisfaction with general oral health	16.17	10.5	−3.368	

**Table 4 ijerph-19-00557-t004:** Open-ended responses of subjects regarding their preference for overdentures.

In Your Opinion, Which is Better (Previous Conventional Dentures or Implant Overdentures) and Why?
Implant overdentures are better because they aid in chewing food.
Implant overdentures are better even though they injure the gums when they are installed and removed.
Implant overdentures are better owing to their better stability.
Implant overdentures are better because conventional dentures are never stable and come out easily from the mouth.
Implant overdentures are better because they are more stable.
Implant overdentures are better because without implants, the dentures would move a lot and hurt the gums and cause ulcers.
Both are almost the same.
Implant overdentures are better because the conventional dentures were moving while I was talking and eating, and I was swallowing food without chewing.
Off course, implant overdentures are better because they are more stable while eating.
Dentures supported by implants are better, in my opinion; a complete removable lower denture should not be made unless it is supported by at least two implants.
Implant overdentures are better because they help with eating, speech, and cleaning.
Dentures without implants are better. My only problem with one of the implants is that it has recently became unstable.
In my experience, implant overdentures are better because they are safer and better.
Implant overdentures are more stable and safer.
Implant overdentures are better. My appearance improved after I started using them. The lips became more supported. My shape became much younger; I am able to go out again and talk to people. Psychologically, I feel better; I am able to eat most types of food.
Implant overdentures are better if both implants are successful.
I do not have problems with the implants; all of them were successful. Although I have diabetes, the operation was successful. My problem is only with the denture itself. It is not stable, and I cannot eat and speak while it is in my mouth.
Implant overdentures are more stable, and I can eat with them.
Implant overdentures are more stable; I did not take advantage of conventional dentures at all.
Implant overdentures are better owing to their stability, ease of eating, and better shape.
Implant overdentures made my oral health better.
Implant overdentures are better because of their stability and better chewing.

## Data Availability

The data are available upon request.

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
