# Peer review of "Patients’ Satisfaction and Oral Health-Related Quality of Life of Edentulous Patients Using Conventional Complete Dentures and Implant-Retained Overdentures in Saudi Arabia"

_ijerph, 2022, doi:10.3390/ijerph19010557_

Round 1
Reviewer 1 Report
Thank you for this helpful contribution. I applaud the effort of promoting studies for the investigation of quality of life. In addition to the below comments, the manuscript would benefit form additional proof reading for grammar and syntax.
Has the study been approved by any ethics committee?
All tables must have the same format.
The conclusion must be clear and concise, it must answer to the main objective.
Author Response
Dear Editor-in-Chief:
I would like to resubmit a revised version of our manuscript entitled: “Patients’ satisfaction and oral health‐related quality of life of edentulous patients using conventional complete dentures and implant‐retained overdentures in Saudi Arabia”.
Here is a point by point, the details of the revisions to the manuscript and our responses to the referees’ comments:
Reply to Reviewer 1:
Thank you so much for the nice comments and valuable enquiries.
In addition to the below comments, the manuscript would benefit form additional proof reading for grammar and syntax.
Manuscript edited by an editing and proofreading service provider.
- Has the study been approved by any ethics committee?
Yes, the study has been approved by the IRB committee at King Saud University “Approval of Research Project No. E-20-4696” and approval letter will be uploaded to the system.
- All tables must have the same format.
Done
- The conclusion must be clear and concise, it must answer to the main objective.
Conclusion was rewritten to be clear and concise and to answer the main objective.
Thanks again

Reviewer 2 Report
I would like to congratulate the authors for this research, and I think it is a good article, and i would like to point out somethings to clarify and help:
1.- There are some mistakes with the editing work, first of all and most important you should edit correctly the bibliography.
In table 1 in the last question the data does not match the lines, in line 61 you should move the satar of the paragraph, in line 129 there is a point out of the paragraph... I suggest you review all the article, but is mandatory to change the bibliography.
2.- The most important thing of the questionnaire is that you take it and it is validated, but you say in the text that you modify it, and if you do so, you should validate your changed questionnnaire, of you don´t do so, you have a no validate questionnnaire. Please review that as well...
Author Response
Thank you so much Sir for the excellent feedback and your rich comments.
- There are some mistakes with the editing work, first of all and most important you should edit correctly the bibliography.
Manuscript edited by an editing and proofreading service provider including the. bibliography.
- In table 1 in the last question the data does not match the lines, in line 61 you should move the satar of the paragraph, in line 129 there is a point out of the paragraph... I suggest you review all the article, but is mandatory to change the bibliography.
All comments addressed and corrected plus Manuscript edited by an editing and proofreading service provider including the. bibliography.
- The most important thing of the questionnaire is that you take it and it is validated, but you say in the text that you modify it, and if you do so, you should validate your changed questionnnaire, of you don´t do so, you have a no validate questionnnaire. Please review that as well...
The questionnaire used was an Arabic translation of the McGill University, 2001 Assessment of Prosthesis which was validated by de Souza et al [20].
de Souza RF, Ribeiro AB, Oates TW, Feine JS. The McGill Denture Satisfaction Questionnaire revisited: Exploratory factor analysis of a binational sample. Gerodontology. 2020 Sep;37(3):233-243. doi: 10.1111/ger.12477. Epub 2020 Jun 3. PMID: 32491236.
Thanks again Sir.

Reviewer 3 Report
The study is well designed and provided using reliable methods.
However, in my opinion, the list of references should be updated, drawing attention to the more modern state of the issue.
I also have a question about a small sample of patients. Is it really possible to draw conclusions about the quality of life in the whole country, whose population is much larger, based on such a number?
Author Response
- Reply to Reviewer 3:
Thank you so much Sir for the excellent feedback and your rich comments.
- The study is well designed and provided using reliable methods.
However, in my opinion, the list of references should be updated, drawing attention to the more modern state of the issue.
This is an excellent and valid comment and we tried to follow up with the most modern and updated references, however, we were obligated to refer to some old but classic papers such as the McGill consensus and the York statement which are very important and provide a concrete base for mandibular implant overdentures.
- Feine J, Carlsson G, Awad M, Chehade A, Duncan WJ, Gizani S et al. The McGill consensus statement on overdentures. Mandibular two-implant overdentures as first choice standard of care for edentulous patients. Gerodontology 2002 Jul;19(1):3-4. doi:10.1111/j.1741-2358.2002.00003.x
- British Society for the Study of Prosthetic Dentistry. The York consensus statement on implant-supported overdentures. Eur J Prosth Restor Dent 2009 Dec;17(4):164-165. PMID: 20158057.
- I also have a question about a small sample of patients. Is it really possible to draw conclusions about the quality of life in the whole country, whose population is much larger, based on such a number?
Thank you for such a valuable question, but in Saudi Arabia, implant overdenture is not as old as it is in western countries particularly in rural areas. We also faced the difficulty of reaching as many patients as we would love to.
Thanks again Sir.

Reviewer 4 Report
Overall, the authors clearly showed that implant-retained lower overdenture significantly improved patients' comfort, aesthetics and general satisfaction over conventional dentures.
The "items" in Table 2 and 3 are suggested to be lowercase to improve readability. And for each "item" in Table 2 is suggested to split into 2 rows, one for conventional denture, the other for implant-overdenture for easier comparison.
Author Response
Reply to Reviewer 4:
Thank you Sir for the valuable comments to make the paper more readable.
Overall, the authors clearly showed that implant-retained lower overdenture significantly improved patients' comfort, aesthetics and general satisfaction over conventional dentures.
- The "items" in Table 2 and 3 are suggested to be lowercase to improve readability.
Done
- And for each "item" in Table 2 is suggested to split into 2 rows, one for conventional denture, the other for implant-overdenture for easier comparison.
Sorry as the table was a bit confusing, I changed the borders and made bold borders for the first column for the conventional denture, I hope this made things clear.
Thanks again and I hope I answered your comments.
Round 2
Reviewer 1 Report
Thanks for the clarifications, this manuscript has been significantly improved.
Reviewer 2 Report
Congratulations, Now the article is ready to be published